# USA vs Russia in the scientific arena

**Giovanni Abramo**[1]*, **Ciriaco Andrea D'Angelo**[2], **Flavia Di Costa**[2]

**1** Laboratory for Studies in Research Evaluation, Institute for System Analysis and Computer Science (IASI-CNR), National Research Council of Italy, Rome, Italy, **2** Department of Engineering and Management, University of Rome "Tor Vergata", Rome, Italy

\* giovanni.abramo@iasi.cnr.it

## Abstract

This work contrasts the scientific standing of the USA and Russia in 146 scientific disciplines. We consider four dimensions of competitive positioning: the contribution to global scientific advancement, the researchers' productivity, the scientific specialization indexes, and the efficiency in resource allocation across disciplines. Differently from previous literature, we use discipline-normalized output to input indicators, thus avoiding distortions due to different intensities of publication across disciplines. Results show that the USA outperforms Russia in contribution to world scholarly impact in all but four disciplines, and is more productive in all but two disciplines. The USA is less efficient in allocating resources to the disciplines where it performs better, probably due to its higher research diversification.

## Introduction

The recent military invasion of Ukraine by the Russian Federation (Russia) has jeopardized an already precarious global geopolitical balance, exacerbating contrasts between Russia and the United States of America (USA), which, together with the bloc of NATO countries, has chosen to support the invaded country through military and other means. The outcomes of such hybrid conflicts are critically reliant on knowledge: both scientific knowledge, embedded in the technologies directly and tangentially functional in the conflict, and economic-managerial knowledge, underlying communications and financial interventions (sanctions, trade restrictions, frozen assets, etc.) aimed at weakening the strategic and economic status of the other side. The scientific and technological standing of nations, i.e. the ability to produce and utilize knowledge effectively and efficiently, is thus not only the main determinant of socio-economic progress, but also a key factor in military conflicts: in decisions to engage or not; in the progress of conflicts engaged, and in the changing standings of nations as the conflicts proceed or end.

In this current work, we aim to compare the competitive strength of Russia and the USA in the scientific arena. Drawing on the microeconomic theory of production and on the theory of international economics, we measure and compare: i) the total scholarly impact of the USA and Russia research activities in 146 disciplines, in the pre-COVID 2015–2019 period; ii) their research productivity, and disciplinary strengths and weaknesses; iii) their respective scientific specializations; and iv) the efficiency of their resource allocations across disciplines.

Following a milestone study by May [1], further scholars have ventured into the complex task of describing and comparing the scientific standing of territories [2–6]. The main

**Data Availability Statement:** All relevant data are within the manuscript and its Supporting information files. In particular, values of bibliometric indicators for all 146 disciplines considered are provided in the Supplementary data S1-S5. Raw data used for calculating indicators

showed in Supporting information files have been extracted from the CWTS in-house WoS database, made available under license by Clarivate Analytics to one of the authors (CAD) as a CWTS research fellow. CWTS should be asked for possible permission to access the same data. Hence, it is not possibile to confirm that "the authors did not have any special access privileges that others would not have", as requested.

**Funding:** The authors received no specific funding for this work.

**Competing interests:** The authors have declared that no competing interests exist.

limitation of this literature, however, is the almost complete omission of any indicators of efficiency (output-to-input) applied to territorial research activities, or at best only at the aggregate level of all disciplines, i.e. subject to significant distortions, failing to account for the differences in knowledge production function across disciplines [7]. For a more profound analysis of previous works and limitations, we refer the reader to Abramo and D'Angelo [8]. What distinguishes the approach of this current study, apart from the very recent and topical observation period, is (i) the application of efficiency indicators obtaining more precise and reliable results than any previous [9]; (ii) the operationalization of new indicators, for more comprehensive representation of reality [10, 11]; and (iii) fine-grained analysis at the disciplinary level, greatly reducing the distortions of analyses at the aggregate level [7]. Future studies might apply the same methodology to compare the scientific standing of other nations.

## Winning the scientific competition

The dimensions of analysis considered representative of the scientific standing of Russia and the USA in each discipline are:

1. the contribution to knowledge advancement, i.e. the world share of the scholarly impact of each country's research activity;

2. the average scholarly impact per researcher, i.e. the productivity of research labor if the capital (all resources other than labor) used is the same;

3. scientific specialization, i.e. the ratio of a country's share of researchers employed in a discipline to the total number of researchers in the country and the analogous ratio worldwide;

4. the efficiency of resource allocation across disciplines.

We assess "who wins" in these dimensions on the basis of four indicators operationally defined in the next Section, serving four functions:

- the measure of the country's contribution to global knowledge advances, which is dependent on both size of research investment and research productivity;

- assessment of the quality of the researchers, thus allowing us to identify a country's disciplines of strength and weakness;

- inquiry into the disciplines in which the country, on average, employs more or less labor than other countries;

- inquiry into the correlation of the country's investments across disciplines with research efficiency.

The analyses begin at the individual level, measuring the value of the indicators for individual researchers, then these results are aggregated at the discipline and country levels. The Materials and Methods section explains how the dataset of national research staff is obtained, and the operationalization of the indicators. In this case, the dataset consists of over 2 million researchers worldwide, of which about 432,000 belong to organizations in the USA and 48,000 in Russia.

For purposes of discipline classification, we use the Web of Science (WoS) scheme, consisting of 254 subject categories (SCs). Each author in the dataset is associated with a discipline given by the "prevalent" SC of their publications. The analysis covers 221 SCs, including all those of the sciences as well as those of the social sciences where WoS coverage is considered representative of research output [12, 13]. For further robustness, we exclude 75 SCs where Russia has less than 10 researchers (of which 26 with no researchers), resulting in a final dataset covering 146 SCs.

## Materials and methods

Analyses concerned the 2015–2019 period and were conducted first at the individual level, i.e. by measuring the value of the bibliometric indicators (see below) in such period for individual researchers, and then at the aggregate level by subject category (SC), area, and country. For the identification of the research staff of countries, we applied the rule-based scoring and clustering algorithm of Caron and van Eck [14] to data extracted from the in-house Web of Science (WoS) database of the Centre for Science and Technology Studies (CWTS) at Leiden University (updated to the 13th week of 2022). For this algorithm, bibliometric metadata on authors and their publications are taken as input, and clusters of publications likely to be written by the same author are taken as output. In order to select the researchers from the two countries, we extracted the clusters featured by an affiliation country "United States" and "Russia". Of course, this excludes from the counting of research staff those researchers who did not publish in journals indexed in WoS.

This algorithm will not be free from error, for example in dealing with authors of very common names (respectively in the USA and Russia), or those with highly diversified and heterogeneous bibliographies, whose portfolios could be split into two or more clusters. However, this latter error would be independent of the authors' country, and so at the aggregate level should have only marginal effects on analytical results.

To exclude "occasional" and no longer active researchers and increase the robustness of the analysis, we exclude those clusters that fail to comply with one or more of the following conditions:

- contain at least 10 publications (excludes "occasional" authors, for whom clustering has lower confidence levels);

- of which at least one publication after 2019 (designed to exclude authors no longer active);

- with a "research age" (given by the difference between the first and the last publication year) of a minimum of 5 years (designed to include only "established" authors).

For field classification purposes, we use the WoS scheme, including 254 subject categories (SCs), falling into 13 areas. Each cluster in the dataset is provided with all related WoS-indexed publications and is associated with one and only one SC, given by the "prevalent" SC of its publications (note that in WoS each publication inherits the SC of the hosting journal). Clusters with more than one prevalent SC are around 7% of researchers, and are randomly assigned one SC among those with a higher frequency.

To avoid distortions due to WoS limitations in the coverage of literature [12, 15, 16] we limit the dataset to all SCs of the sciences and several SCs of the social sciences. Among these, we further exclude 75 SCs where Russia has less than 10 researchers (26 of these with no researchers at all), The field of observation then includes 146 SCs grouped in 11 areas; the final dataset consists of over 2 million authors, of which 432,000 for the USA (ranking first in the world) and 48,000 for Russia.

Fig 1 illustrates the workflow for the data selection and refinement, while Table 1 shows the breakdown of clusters by area for the two countries.

The dimensions of analysis considered to represent the scientific standing of the two countries are four and are measured through four indicators:

- The first (which in the following we will refer to as TFI, or total fractional impact) measures the relative contribution of a country to global knowledge advances.

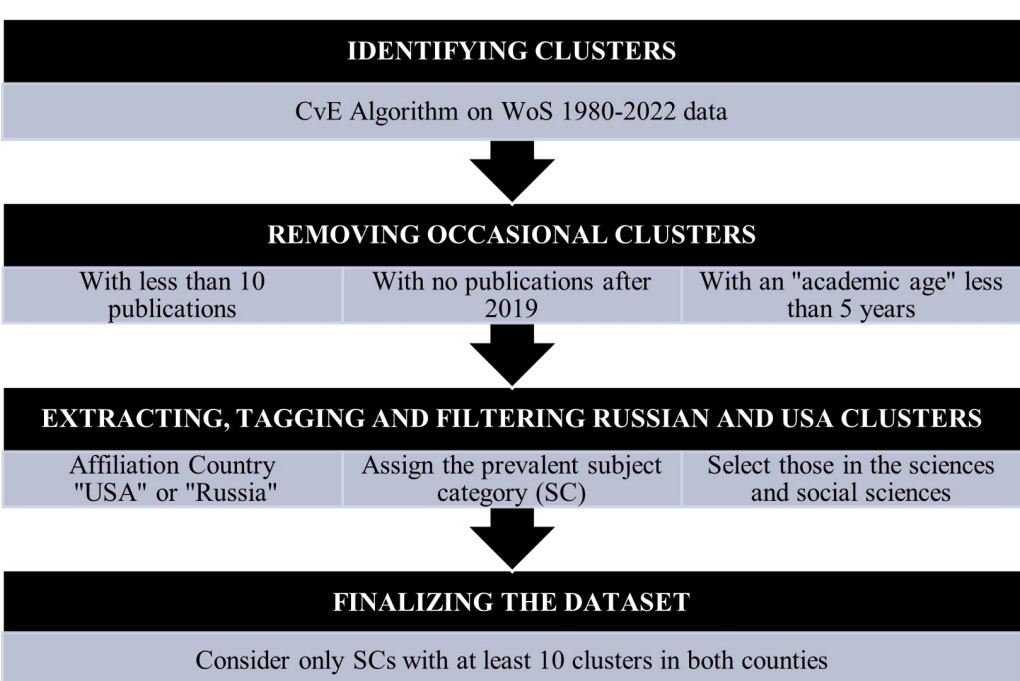

**Fig 1. Workflow of the dataset construction.**

- The second (hereafter referred to as FSS, or fractional scientific strength) reveals how good researchers are at research, and also allows us to identify a country's SCs of strength and weakness.

- The third (which we will refer to in the following as SI, or specialization index) reveals the SCs in which the country invests more or less than other countries on average.

- The fourth measures the correlation between specialization (SI) and productivity (FSS) across SCs, revealing whether the country invests more in the SCs in which it is most productive.

**Table 1. Number of clusters by country and area.**

| Area | Russia | USA | World | Share Russia/USA | Share USA/World |
|---|---|---|---|---|---|
| Biology | 6027 | 67250 | 317982 | 9.0% | 21.1% |
| Biomedical research | 2617 | 76699 | 294088 | 3.4% | 26.1% |
| Chemistry | 7560 | 20996 | 144063 | 36.0% | 14.6% |
| Clinical medicine | 3919 | 138227 | 520188 | 2.8% | 26.6% |
| Earth and space sciences | 3236 | 20291 | 120273 | 15.9% | 16.9% |
| Economics | 198 | 8323 | 30402 | 2.4% | 27.4% |
| Engineering | 8225 | 46472 | 367770 | 17.7% | 12.6% |
| Political and social sciences | 369 | 9795 | 28950 | 3.8% | 33.8% |
| Mathematics | 1391 | 7958 | 45815 | 17.5% | 17.4% |
| Physics | 14322 | 33685 | 203012 | 42.5% | 16.6% |
| Psychology | 63 | 2677 | 7396 | 2.4% | 36.2% |
| Total | 47927 | 432373 | 2079939 | 11.1% | 20.8% |

A working definition is provided below.

## The total research impact of a country

For each SC, the relative scientific weight of a country is measured through a composite output-side indicator, called Total Fractional Impact (TFI). The TFI of a country $k$ in a SC $j$, is defined as:

$$TFI_{jk} = \sum_{i=1}^{N_{jk}} f_{ik} \frac{c_i}{\bar{c}_j} \qquad (1)$$

$N_{jk}$ = number of 2015–2019 WoS publications of authors in SC $j$ of country $k$.

$f_{ik}$ = fractional contribution of co-authors of country $k$ to publication $i$. For a publication with $n$ co-authors, $m$ of which affiliated to country $k$, $f_{ik}$ is equal to $m/n$.

$c_i$ = citations received by publication $i$ (counted at the 13th week of 2022).

$\bar{c}_j$ = average citations received by all cited publications of the same year and SC $j$ of publication $i$.

In brief, TFI is calculated at the country level through the standardized citations received from the publications produced in the observation period (the number of citations rescaled by year and SC, thereby avoiding distortions in comparing publications of different years and SC) multiplied by the fractional contribution of co-authors of country $k$ to each publication.

Once TFI is calculated, a country's relative scientific impact is simply obtained by dividing the value recorded for that country by the world total.

## Research productivity of a country

We measure the scientific productivity of a country starting from the individual level through the Fractional Scientific Strength indicator or $FSS_p$, defined as:

$$FSS_p = \sum_{i=1}^{N} \frac{c_i}{\bar{c}_j} f_i \qquad (2)$$

where:

$N$ = number of WoS publications by the author in the period under observation.

$f_i$ = fractional contribution of the author to publication $i$ (given by the reciprocal of the number of coauthors in the byline).

The productivity of countries, which are heterogeneous in the research fields of their staff, cannot be directly measured at the aggregate level. So, after measuring the productivity of individual authors (Eq 2) we normalize individual productivity by the average of the relevant SC world distribution. At the aggregate level then, the yearly productivity $FSS_A$ for the aggregate unit $A$ is:

$$FSS_A = \frac{1}{RS} \sum_{j=1}^{RS} \frac{FSS_{P_j}}{\overline{FSS_P}} \qquad (3)$$

Where:

$RS$ = number of authors in the unit, in the observed period;

$FSS_{P_j}$ = productivity of author $j$ in the unit;

$\overline{FSS_P}$ = average productivity of all world authors in the same SC of author $j$.

A value of $FSS_A$ = 1.20 means that the country unit A employs authors with average productivity of 20% higher than expected, i.e. the world average.

In this way, we can measure the productivity of a country at SC, area, and overall level, avoiding distortions due to the different intensities of publication and citation across SCs.

A thorough description of the methodology, assumptions and limitations, underlying theory and data source can be found in Abramo and D'Angelo [17].

## Research specializations of a country

To assess the degree of specialization of a country in each SC, we use the Balassa index applied to input data [18], revealing whether a country specializes in a specific SC relative to other countries. Named $RS$ the total number of authors of country $i$ in SC $j$, the specialization index $SI_{ji}$ is:

$$SI_{ji} = \frac{RS_{ji}}{\sum_j RS_{ji}} \bigg/ \frac{\sum_i RS_{ji}}{\sum_j \sum_i RS_{ji}} \tag{4}$$

The higher the value of $SI_{ji}$ compared to one, the more specialized the country $i$ is in SC $j$. If $SI_{ji}$ is less than one, it means that no specialization is involved in $j$ for country $i$.

## Efficiency in resource allocation

This dimension concerns the alignment between comparative advantage (FSS) and scientific specialization (SI) across SCs for a given country. In other words, whether and how much a country concentrates its researchers in the SCs in which it is most productive. The measure used for this purpose is Kendall's tau-b correlation between the FSS and SI values recorded in the 146 SCs considered in the study.

The data and analyses are reported in detail in S1 to S5 in S1 Data. In the next section, we summarize some significant results.

# Results

## Scholarly impact of research activities

The ratio of the USA to Russia research staff is 9:1, and in favor of the USA in all but nine SCs, a staffing imbalance inevitably reflected in overall scientific outcomes. The distribution of the share of scholarly impact (TFI) with respect to the world total, for the 146 SCs analyzed and reported in Table 2 at an aggregate level of 11 areas, presents a median value of 23.6% for the USA and 0.4% for Russia (averages respectively 24.3% and 1.4%). The USA has the highest global TFI share in 94 (64%) SCs, and ranks second in another 36 (25%). In 107 (73%) SCs the USA TFI is 10 times higher than for Russia; in 90 SCs by 20 times; in 71 by 50; and in 51 by 100. The USA produces more than half of the world's TFI in five SCs: Psychology; Pathology; Engineering, petroleum; Sociology; and Biophysics.

In contrast, Russia does not excel in any of the 146 SCs considered and outperforms the USA in only 4 SCs. Russia's best ranking is in 'Crystallography', where it attains 7.1% of world TFI, behind only China (17.4%) and India (8.2%), and slightly ahead of the USA (6.8%). Russia leads the USA in only three other SCs: Mineralogy (9.4% vs 7.8%), Mining & mineral processing (6.8% vs 6.4%), and Chemistry, inorganic & nuclear (6.0% vs 4.1%).

**Table 2. Descriptive statistics of the share of world scholarly impact (TFI) by discipline reported at the research area level.**

| Area | No. of SCs | Russia | | | USA | | |
|---|---|---|---|---|---|---|---|
| | | Min | Max | Average | Min | Max | Average |
| Biology | 21 | 0.0% | 3.2% | 0.8% | 7.4% | 50.9% | 25.3% |
| Biomedical research | 12 | 0.1% | 3.1% | 0.5% | 14.4% | 55.1% | 32.7% |
| Chemistry | 8 | 1.1% | 6.0% | 2.7% | 4.1% | 26.9% | 10.6% |
| Clinical medicine | 24 | 0.0% | 0.8% | 0.2% | 8.3% | 43.3% | 32.6% |
| Earth and space sciences | 11 | 0.1% | 9.4% | 2.1% | 2.6% | 38.0% | 19.0% |
| Economics | 3 | 0.3% | 0.4% | 0.4% | 28.3% | 36.4% | 31.0% |
| Engineering | 35 | 0.1% | 10.9% | 1.4% | 4.5% | 53.0% | 17.7% |
| Mathematics | 5 | 0.3% | 4.1% | 1.4% | 14.1% | 35.1% | 21.7% |
| Physics | 17 | 0.5% | 7.3% | 4.0% | 6.8% | 34.8% | 18.3% |
| Psychology | 3 | 0.1% | 0.2% | 0.2% | 35.7% | 57.7% | 45.7% |
| Social sciences | 7 | 0.0% | 1.5% | 0.5% | 26.1% | 52.1% | 39.4% |
| Total | 146 | 0.0% | 10.9% | 1.4% | 2.6% | 57.7% | 24.3% |

## Research productivity

In five SCs (Physics, condensed matter; Materials science, characterization & testing; Metallurgy & metallurgical engineering; Chemistry, applied; Material science, ceramics) Russia has a larger research staff than the USA but a lower TFI: this means that the USA researchers must be more productive, and/or that in these five SCs, Russia provides less capital investment. In all other SCs, one of the countries registers higher in both TFI and research staff, and we cannot immediately obtain inferences on relative research productivity. Instead, measuring for each individual SC, Fig 2 shows the box plot of the fractional scientific strength (FSS) per researcher, normalized to the world average (e.g. FSS = 1.20 means that the researchers average a scholarly impact 20% higher than the average of world researchers in the same SC). American researchers outperform Russian researchers in all but two SCs: Engineering, aerospace, where 111 Russian researchers reach an average FSS of 0.806, greater than 0.786 for 966 American researchers; Entomology, where 49 Russian researchers achieve an average FSS of 1.589, greater than the 1.334 of the 1835 American researchers. In all other 144 SCs, Americans achieve average FSS scores significantly higher than those for the Russians.

In particular, the average FSS of American researchers is higher than the world average in 121 (83%) SCs; Russia achieves this in only two SCs (Spectroscopy, and Entomology). Fig 3 shows the data aggregated for the 146 SCs by area: American superiority is apparent in all areas, with a maximum delta in Psychology (+685%), followed by Clinical medicine (+434%) and Biomedical research (+303%), and a minimum in Mathematics (+54%). Overall, the FSS of American researchers is three times that of Russian colleagues (1.213 vs 0.410).

## Scientific specializations

The two countries show very different profiles in disciplinary specialization: Russia tends to concentrate high numbers of researchers in a limited number of SCs; the USA has research staff distributed much more evenly among SCs. Across the 146 SCs, the standard deviation of the specialization index (the higher the SI, the more specialized the country is in that discipline) is 1.682 for Russia, as opposed to 0.507 for the USA; the GINI coefficient of distribution (the most commonly used measure of inequality, varying between 0 in case of complete equality and 1, indicating complete inequality), is respectively 0.587 and 0.291 (0 expresses perfect equality; 1 expresses maximal inequality).

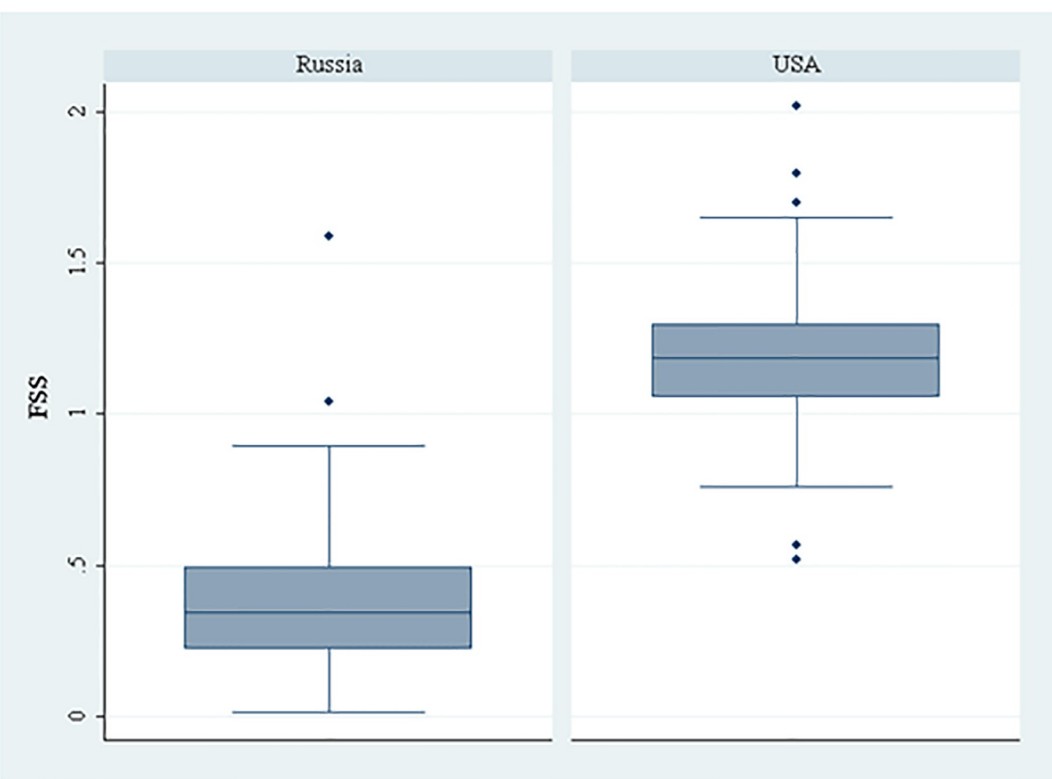

**Fig 2. Boxplot of research productivity (FSS) in the 146 disciplines (SCs) under investigation, by country.**

The USA shows only four SCs with SI < 0.2; and only seven with SI > 2. In contrast, Russia presents 24 SCs with SI < 0.2 and 34 SCs with SI > 2. Recall that for Russia, the dataset excluded a full 75 SCs with fewer than 10 researchers, including 26 with no researchers at all. With inclusion of these, the observed level of concentration would be still greater.

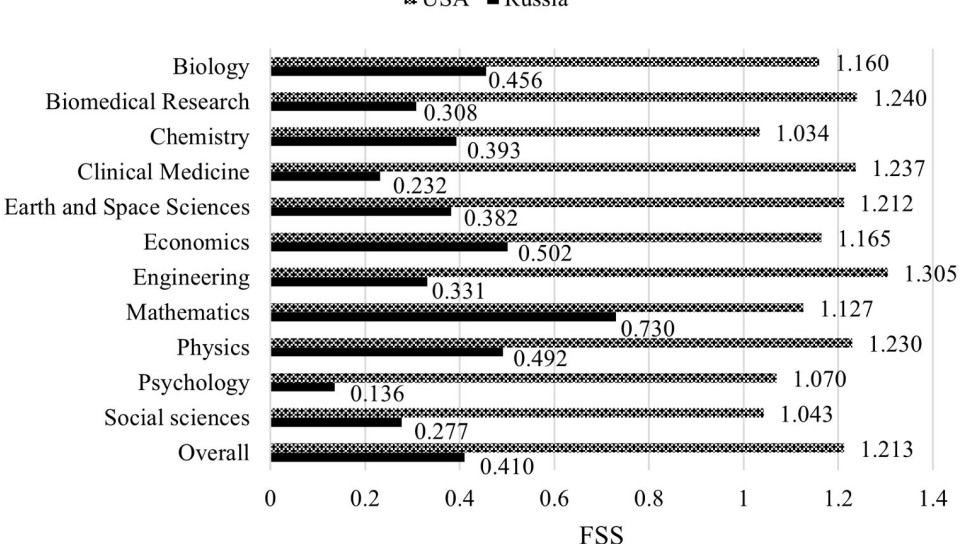

**Fig 3. Research productivity of the two countries, by area and overall.**

**Table 3. Disciplines (SCs) with the maximum (top half) and the minimum (bottom half) differences between specialization indexes (SI) of the two countries.**

| Discipline | Area | SI_Russia | SI_USA |
|---|---|---|---|
| Materials Science, Characterization & Testing | Engineering | 8.39 | 0.40 |
| Mining & Mineral Processing | Engineering | 7.15 | 0.21 |
| Chemistry, Applied | Chemistry | 6.81 | 0.20 |
| Physics, Condensed Matter | Physics | 6.85 | 0.38 |
| Engineering, Petroleum | Engineering | 7.95 | 1.48 |
| Metallurgy & Metallurgical Engineering | Engineering | 5.87 | 0.19 |
| Mineralogy | Earth and Space Sciences | 5.22 | 0.30 |
| Crystallography | Physics | 5.07 | 0.20 |
| Chemistry, Inorganic & Nuclear | Chemistry | 4.33 | 0.24 |
| Materials Science, Ceramics | Engineering | 4.34 | 0.28 |
| Information Science & Library Science | Engineering | 0.27 | 1.71 |
| Health Policy & Services | Social sciences | 0.28 | 1.77 |
| Public, Environmental & Occupational Health | Clinical Medicine | 0.06 | 1.62 |
| Sport Sciences | Clinical Medicine | 0.04 | 1.61 |
| Anthropology | Social sciences | 0.41 | 2.08 |
| Pathology | Biomedical Research | 0.24 | 1.96 |
| Critical Care Medicine | Clinical Medicine | 0.07 | 1.84 |
| Psychology | Psychology | 0.63 | 2.43 |
| Health Care Sciences & Services | Clinical Medicine | 0.06 | 2.09 |
| Political Science | Social sciences | 0.10 | 2.19 |

For Russia, the ten SCs of highest specialization are mostly in Engineering, i.e.: Materials science, characterization & testing (8.4); Engineering, petroleum (7.9); Mining & mineral processing (7.2); Physics, condensed matter (6.9); Chemistry, applied (6.8); Metallurgy & metallurgical engineering (5.9); Mineralogy (5.2); Crystallography (5.1); Materials science, ceramics (4.3); Chemistry, inorganic & nuclear (4.3).

By contrast, the USA shows higher diversification, with SI of the SCs in which it specializes generally around 2: Psychology (2.4); Psychology, educational (2.4); Biophysics (2.3); Sociology (2.2); Political science (2.2); Health care sciences & services (2.1); Anthropology (2.1); Pathology (2.0); Critical care medicine (1.8); Health policy & services (1.8).

Table 3 helps characterize the substantial differences in specialization profiles between Russia and the USA, by listing the 10 SCs with the greatest and smallest differences for this country pair.

Overall, Russia is more specialized than the USA in 63 (43%) SCs. In particular, Russia has specialization indexes higher than the USA in every SC of Chemistry and of Physics; in Earth and space sciences this happens in nine of total 11 SCs. However, in five other areas (Social sciences, Biomedical research, Clinical medicine, Economics, Psychology) comprising a total 49 SCs, Russian specialization is higher than American in only four of them.

## Efficiency in resource allocation

Comparing the two countries, we can now inquire into the alignment across SCs between comparative advantage (FSS) and scientific specialization (SI). In other words, whether and to what extent Russia and the USA concentrate their researchers in the SCs where the country is most productive. For this, we use Kendall's tau-b correlation between the FSS and SI values for the 146 SCs: the results indicate that for Russia there is a weak but significant correlation

(Kendall's tau-b 0.200, p-value 0.0003) between the two dimensions of analysis; instead for the USA this correlation is actually negative in sign (Kendall's tau-b -0.095) and with p-value 0.0894, therefore practically absent.

In spite of lower efficiency in cross-SC resource allocation, possibly due to strategies of greater diversification, the USA unequivocally outperforms Russia in research productivity and in contributing to world knowledge advancement. In one-third of the SCs considered, the USA scholarly impact is at least 100 times greater than that of Russia; and in all but two SCs, American researchers achieve higher scholarly impact.

The strong science basis observed for the USA leaves aside the questions of national ability in incorporating new knowledge in products, services, and operations, leading to economic strength. In this area too, however, American abilities in technology transfer have long been recognized [19].

Looking at knowledge advances emerging from the USA and Russia, it could be interesting to focus new research on the collaboration networks and knowledge flows between the two countries and with the rest of the world.

## Conclusions

In 2019, before the COVID-19 pandemic outbreak, the USA and Russia presented marked differences in their economic profiles [20]. The OECD official statistics report that the USA, with a population of about 328 Mln persons, registered a GDP per capita (65,081 USD current PPPs) [21], more than twice that of Russia (29,967 USD current PPPs), whose population was around 147 Mln persons [22]. In the USA, there were 9.9 full-time equivalent researchers per '000 employed, and gross expenditure on research was 632,655 Mln USD, against respectively 5.6 and 36,201 Mln USD in Russia.

The research expenditure and staff imbalance reflect the differing contribution to the advances of science. The analyses on the scholarly impact and scientific specialization by field show an alignment between economic and scientific specialization, confirming the theory of comparative advantage. Russia's economic comparative advantage in natural resources pulls its outstanding scientific specialization and contribution to advance the frontiers of knowledge, especially in such fields as materials science, petroleum and metallurgic engineering, mineralogy, and the like. Such findings might work as a timely antidote to the often-heard superficial talk about Russia as a country dependent on natural resources, ignoring its important scientific contribution to the fields.

The USA is not just larger in research size but also outperforms Russia in labor research productivity, in all fields but entomology and aerospace engineering, with the latter confirming that the cold war contest in space exploration between the two countries is still on. Space exploration nowadays embeds more and more scientific experiments, especially in biology and medical sciences. The extent of the revealed labor productivity gap might be partly explained by the employment of more capital in research activities by the USA, and the low coverage of Russian journals by WoS.

The two countries differ also in their research diversification strategies, with the USA being highly diversified and Russia much more concentrated. While the USA higher diversification allows capturing returns to scope, where potentially available, on the other hand, they might partly be a cause of lower efficiency in funds allocation across fields, as findings reveal.

When interpreting the results of the proposed analyses, all assumptions and limits of the bibliometric methods apply. First, not all new knowledge produced is encoded in publications in scientific journals, and not all scientific journals are indexed in bibliographic repertories. Furthermore, measuring the value of publications using citation-based indicators is a

prediction, not definitive. Also, citations can be negative or inappropriate; in any case, they certify only scholarly impact, forgoing other types of impact. It should also be noted that the analytical results would vary with the selection of a different field classification scheme. But probably the most significant aspect, given the objectives of the proposed analysis, is the imbalance of scientific journals covered by WoS: according to Clarivate, the 2021 Journal Citation Index covers 5930 journals published in the USA compared to 390 in Russia. Since roughly 2010, Russian researchers have had strong and increasing incentives to publish in indexed journals. However, there could be many who still publish in non-indexed, solely Russian-language journals [23, 24].

## Supporting information

**S1 Data.**
(XLSX)

**S2 Data.**
(PDF)

## Acknowledgments

We thank the Centre for Science and Technology Studies (CWTS) at Leiden University for providing us with access to the in-house WoS database from which we extracted data at the basis of our elaborations.

## Author Contributions

**Conceptualization:** Giovanni Abramo, Ciriaco Andrea D'Angelo.

**Data curation:** Ciriaco Andrea D'Angelo, Flavia Di Costa.

**Investigation:** Giovanni Abramo, Ciriaco Andrea D'Angelo, Flavia Di Costa.

**Methodology:** Giovanni Abramo, Ciriaco Andrea D'Angelo, Flavia Di Costa.

**Supervision:** Giovanni Abramo, Ciriaco Andrea D'Angelo.

**Validation:** Ciriaco Andrea D'Angelo.

**Visualization:** Flavia Di Costa.

**Writing – original draft:** Giovanni Abramo, Ciriaco Andrea D'Angelo, Flavia Di Costa.

**Writing – review & editing:** Giovanni Abramo, Ciriaco Andrea D'Angelo.

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
