## [Decision Letter · Decision Letter 0]

2 Feb 2023

PONE-D-22-31771

USA vs Russia in the scientific arena

PLOS ONE

Dear Dr. Abramo,

Thank you for submitting your manuscript to PLOS ONE. After careful consideration, we feel that it has merit but does not fully meet PLOS ONE’s publication criteria as it currently stands. Therefore, we invite you to submit a revised version of the manuscript that addresses the points raised during the review process.

We look forward to receiving your revised manuscript.

Kind regards,

Bruno Miguel Pinto Damásio

Academic Editor

PLOS ONE

Journal Requirements:

Upon re-submitting your revised manuscript, please upload your study’s minimal underlying data set as either Supporting Information files or to a stable, public repository and include the relevant URLs, DOIs, or accession numbers within your revised cover letter. For a list of acceptable repositories, please see http://journals.plos.org/plosone/s/data-availability#loc-recommended-repositories. Any potentially identifying patient information must be fully anonymized

Additional Editor Comments:

Dear Dr Giovanni Abramo

Thank you for considering PLOS ONE for submitting your article.

I already have the referee reports from three reviewers and I am pleased to inform you that their appreciation of your manuscript is positive. Mine is also, you have here a quality work worthy of being published in PLOS ONE.

However, the authors point out several points that need to be improved/clarified in your manuscript before it is ready for publication.

So I am happy to allow you to submit a new and improved version.

Best regards,

Bruno Damásio.

Reviewers' comments:

Reviewer's Responses to Questions

**Comments to the Author**

1. Is the manuscript technically sound, and do the data support the conclusions?

Reviewer #1: Yes

Reviewer #2: Yes

Reviewer #3: Yes

2. Has the statistical analysis been performed appropriately and rigorously? 

Reviewer #1: Yes

Reviewer #2: Yes

Reviewer #3: Yes

3. Have the authors made all data underlying the findings in their manuscript fully available?

Reviewer #1: Yes

Reviewer #2: Yes

Reviewer #3: Yes

4. Is the manuscript presented in an intelligible fashion and written in standard English?

Reviewer #1: Yes

Reviewer #2: Yes

Reviewer #3: Yes

5. Review Comments to the Author

Reviewer #1: This paper is well done, it is brave in its object of analysis, and it achieves its goals with a conscientious methodology. I think the paper should not be published before a number of (minor) issues be attended to.

Some notes:

p. 3 – basic details about sources/metrics/indicators could perhaps be nailed in a minimalist but effective section that can help the reader in moving forward without having to jump to the “Materials” section below …

p. 3 – in the “Results” section you refer to science staff USA vs RU is 9:1. All right, however:

- you should report the source (but note these data is not coming from WoS, so where does it come from?) and put out a word regarding comparability (the science systems are quite different; plus, there devilish details, e.g., how is university staff counted in these statistics? how are corporate researchers counted? etc.)

- could you situate the reader regarding the relative sizes of the country populations, GDPs, absolute R&D figures? … note that the US is about 14x the size of the ruskie economy … this book is old, but gives some comparative perspective https://books.google.pt/books?hl=pt-PT&lr=&id=T4doAgAAQBAJ&oi=fnd&pg=PP1&dq=size+of+US+and+russian+economies&ots=7SHXOBUnMt&sig=GSKBDHKhZcP-QT7scNTdjCZNiuo&redir_esc=y#v=onepage&q=size%20of%20US%20and%20russian%20economies&f=false

p. 4 – Regarding research productivity: we perhaps should not forget that US researchers publish directly in the global language (today’s latin) and that they control the outlets (see the recently growing literature on “editormetrics”, e.g. https://link.springer.com/article/10.1007/s11192-022-04279-9 and https://link.springer.com/article/10.1007/s11192-017-2434-7)

p. 5 – for sure in terms of efficiency something of the kind seems to be going on in terms of military spending … the US military-industrial complex is too expensive (costs in excess to the original procurement contracts, one thinks of the F35) for the results it delivers on the battle ground while russian gear appears to be quite sturdy (lighter in logistics, less needed of repairs) … see https://link.springer.com/chapter/10.1057/9781403983428_8

p. 6 – I feel it would be useful to have a table summarising the key findings and highlighting the key knowledge fields that were reported. For instance, this paper provides an important and timely antidote to the superficial talk on Russia as a country dependent on natural resources (science indicators show that this country does a lot to develop frontier knowledge about minerals, etc.). Any changes that the authors could do to favour the papers usability would be welcome and would leverage the impact of this work, which I would wish should be big.

Note: I wonder if this paper would not be useful in any way

https://direct.mit.edu/qss/article/3/1/37/109076/Scopus-1900-2020-Growth-in-articles-abstracts

Reviewer #2: Summary

This work compares the scientific output between USA and Russia in 146 scientific disciplines. The results show that USA outperforms Russia in world impact. However, USA is less efficient in allocating resources to the disciplines where it performs better.

Some general comments

The "Materials and Methods" section should be before the "Winning the competition/Results" after the "Introduction" section. You are mentioning and analyzing relevant measures in your methodology, such as FSS, and only explain what they are at the end of the article. Also, there should be a section with Conclusions/Discussion regarding the work developed.

The "Materials and Methods" should have a diagram workflow for the data selection and refinement. Also, specify the period considered (2015-2019) in this diagram/section.

In Figure 1, the x-axis should have a label. From the text, it is easy to understand that each point represents one of the 146 of the SC, but by analyzing the plot, it is not straightforward. Also, in that case, you aim to plot a categorical variable against a continuous variable. The most appropriate graph for that is not a scatter plot, but a side-by-side boxplot. Since you have a lot of SC categories and areas, you could show the side-by-side for the overall measures and comment on the differences between areas in the text.

Additionally, the information in Table 2 can be represented through a barplot (x- area and y-FSS) per country. The same thing applies to Table 1.

Reviewer #3: The topic is actually interesting and developed at a time when everyone is looking for topics about Russia. The research is well-described, and the findings are extremely useful. For these reasons, I believe the manuscript can be published, but it requires major revisions and reorganization.

It surprised me to see the results section before the materials and methods section. That should be changed so that readers can better understand the results. For example, the results discuss the "distribution of share of scholarly impact" and refer to TFI in brackets. However, it has not yet been defined, making it difficult to comprehend.

All previously used and defined short expression forms are defined again in the material and methods section. SC, for example, is defined twice in this section. Furthermore, "The field of observation then includes 146 SCs grouped in 11 areas" appears to be a new sentence that should follow a full stop.

Although the Gini coefficient is well known, it would benefit from a definition or reference.

Finally, I haven't come across any discussion or conclusion sections, and the paper would benefit from them.

6. PLOS authors have the option to publish the peer review history of their article (what does this mean?). If published, this will include your full peer review and any attached files.

Reviewer #1: No

Reviewer #2: No

Reviewer #3: No

---

## [Author Response · Author response to Decision Letter 0]

28 Feb 2023

Reviewer #1

This paper is well done, it is brave in its object of analysis, and it achieves its goals with a conscientious methodology. I think the paper should not be published before a number of (minor) issues be attended to.

Reply by the authors

We thank the reviewer for his appreciation. We tried to follow up on all his comments and revise the manuscript accordingly.

Some notes:

p. 3 – basic details about sources/metrics/indicators could perhaps be nailed in a minimalist but effective section that can help the reader in moving forward without having to jump to the “Materials” section below …

Reply by the authors

We have anticipated the Materials and Methods section.

p. 3 – in the “Results” section you refer to science staff USA vs RU is 9:1. All right, however:

- you should report the source (but note these data is not coming from WoS, so where does it come from?) and put out a word regarding comparability (the science systems are quite different; plus, there devilish details, e.g., how is university staff counted in these statistics? how are corporate researchers counted? etc.)

Reply by the authors

By research staff we mean the authors of publications indexed in WoS. The methodology to identify those authors is presented at the very beginning of the Materials and Methods section. We added a footnote to make it more clear: “The adopted methodology excludes from the counting of research staff researchers who did not publish in journals indexed in WoS.” To conduct our analyses we need names and field of research of each scientist. No databases containing that information are available. Consequently, we had to devise the above said methodology to extract the data.

- could you situate the reader regarding the relative sizes of the country populations, GDPs, absolute R&D figures? … note that the US is about 14x the size of the ruskie economy … this book is old, but gives some comparative perspective https://books.google.pt/books?hl=pt-PT&lr=&id=T4doAgAAQBAJ&oi=fnd&pg=PP1&dq=size+of+US+and+russian+economies&ots=7SHXOBUnMt&sig=GSKBDHKhZcP-QT7scNTdjCZNiuo&redir_esc=y#v=onepage&q=size%20of%20US%20and%20russian%20economies&f=false

Reply by the authors

We reported a few statistics about the two countries in the Conclusions section.

p. 4 – Regarding research productivity: we perhaps should not forget that US researchers publish directly in the global language (today’s latin) and that they control the outlets (see the recently growing literature on “editormetrics”, e.g. https://link.springer.com/article/10.1007/s11192-022-04279-9 and https://link.springer.com/article/10.1007/s11192-017-2434-7)

Reply by the authors

We agree. In fact, in the Conclusions section we state that “the 2021 Journal Citation Index covers 5930 journals published in the USA compared to 390 in Russia. Since roughly 2010, Russian researchers have had strong and increasing incentives to publish in indexed journals. However, there could be many who still publish in non-indexed, solely Russian-language journals.”

p. 5 – for sure in terms of efficiency something of the kind seems to be going on in terms of military spending … the US military-industrial complex is too expensive (costs in excess to the original procurement contracts, one thinks of the F35) for the results it delivers on the battle ground while russian gear appears to be quite sturdy (lighter in logistics, less needed of repairs) … see https://link.springer.com/chapter/10.1057/9781403983428_8

Reply by the authors

We trust the reviewer but, frankly speaking, we are not competent in military issues. The suggested publication dates 2006 and we do not know whether any changes have occurred ever since. We prefer to avoid entering a field we are not competent in.

p. 6 – I feel it would be useful to have a table summarising the key findings and highlighting the key knowledge fields that were reported. For instance, this papefr provides an important and timely antidote to the superficial talk on Russia as a country dependent on natural resources (science indicators show that this country does a lot to develop frontier knowledge about minerals, etc.). Any changes that the authors could do to favour the papers usability would be welcome and would leverage the impact of this work, which I would wish should be big.

Note: I wonder if this paper would not be useful in any way

https://direct.mit.edu/qss/article/3/1/37/109076/Scopus-1900-2020-Growth-in-articles-abstracts

Reply by the authors

We have integrated the paper with a concluding section where we wrap up the main findings and possible interpretations. We are somewhat hesitant to extend further their implications, as we are essentially bibliometricians, with no competencies in international policies and the like.

Reviewer #2

This work compares the scientific output between USA and Russia in 146 scientific disciplines. The results show that USA outperforms Russia in world impact. However, USA is less efficient in allocating resources to the disciplines where it performs better.

Some general comments

Reply by the authors

We wish to thank the reviewer for his valuable comments and suggestions. Our detailed feedback to each individual comment follows.

The "Materials and Methods" section should be before the "Winning the competition/Results" after the "Introduction" section. You are mentioning and analyzing relevant measures in your methodology, such as FSS, and only explain what they are at the end of the article. Also, there should be a section with Conclusions/Discussion regarding the work developed.

Reply by the authors

We have anticipated the Materials and methods section and integrated the paper with a concluding section.

The "Materials and Methods" should have a diagram workflow for the data selection and refinement. Also, specify the period considered (2015-2019) in this diagram/section.

Reply by the authors

Done

In Figure 1, the x-axis should have a label. From the text, it is easy to understand that each point represents one of the 146 of the SC, but by analyzing the plot, it is not straightforward. Also, in that case, you aim to plot a categorical variable against a continuous variable. The most appropriate graph for that is not a scatter plot, but a side-by-side boxplot. Since you have a lot of SC categories and areas, you could show the side-by-side for the overall measures and comment on the differences between areas in the text.

Reply by the authors

Done

Additionally, the information in Table 2 can be represented through a barplot (x- area and y-FSS) per country. The same thing applies to Table 1.

Reply by the authors

We agree with the reviewer’s suggestion concerning Table 2 (turned into Table 3 after the revision), but not for Table 1 since it shows different descriptive statistics that would be difficult to condense into a single figure.

Reviewer #3

The topic is actually interesting and developed at a time when everyone is looking for topics about Russia. The research is well-described, and the findings are extremely useful. For these reasons, I believe the manuscript can be published, but it requires major revisions and reorganization.

Reply by the authors

We thank the reviewer for his appreciation. We tried to follow up on all his comments and revise the manuscript accordingly.

It surprised me to see the results section before the materials and methods section. That should be changed so that readers can better understand the results. For example, the results discuss the "distribution of share of scholarly impact" and refer to TFI in brackets. However, it has not yet been defined, making it difficult to comprehend.

All previously used and defined short expression forms are defined again in the material and methods section. SC, for example, is defined twice in this section. Furthermore, "The field of observation then includes 146 SCs grouped in 11 areas" appears to be a new sentence that should follow a full stop.

Reply by the authors

We have solved this, anticipating the Materials and methods section.

Although the Gini coefficient is well known, it would benefit from a definition or reference.

Reply by the authors

We have defined the Gini coefficient.

Finally, I haven't come across any discussion or conclusion sections, and the paper would benefit from them.

Reply by the authors

We have integrated the paper with a concluding section.

---

## [Decision Letter · Decision Letter 1]

21 Jun 2023

USA vs Russia in the scientific arena

PONE-D-22-31771R1

Dear Dr. Abramo,

We’re pleased to inform you that your manuscript has been judged scientifically suitable for publication and will be formally accepted for publication once it meets all outstanding technical requirements.

Kind regards,

Bruno Miguel Pinto Damásio

Academic Editor

PLOS ONE

Additional Editor Comments (optional):

Dear Dr Giovanni Abramo,

I am very pleased to inform you that your paper has been accepted for publication in PLOS ONE.

Your article is original, innovative, and a very relevant piece of research. Congratulations!

Best regards,

Bruno Damásio

Reviewers' comments:

Reviewer's Responses to Questions

**Comments to the Author**

1. If the authors have adequately addressed your comments raised in a previous round of review and you feel that this manuscript is now acceptable for publication, you may indicate that here to bypass the “Comments to the Author” section, enter your conflict of interest statement in the “Confidential to Editor” section, and submit your "Accept" recommendation.

Reviewer #1: (No Response)

Reviewer #2: All comments have been addressed

2. Is the manuscript technically sound, and do the data support the conclusions?

Reviewer #1: Yes

Reviewer #2: Yes

3. Has the statistical analysis been performed appropriately and rigorously? 

Reviewer #1: Yes

Reviewer #2: Yes

4. Have the authors made all data underlying the findings in their manuscript fully available?

Reviewer #1: Yes

Reviewer #2: Yes

5. Is the manuscript presented in an intelligible fashion and written in standard English?

Reviewer #1: Yes

Reviewer #2: Yes

6. Review Comments to the Author

Reviewer #1: Final comments:

p. 3 - "terminated researchers" ?!

p. 4 - "CvE"?

Bibliography - Diana Hicks appears two times in ref 13 and 17

Reviewer #2: (No Response)

7. PLOS authors have the option to publish the peer review history of their article (what does this mean?). If published, this will include your full peer review and any attached files.

Reviewer #1: No

Reviewer #2: No

---

## [Editor Report · Acceptance letter]

23 Jun 2023

PONE-D-22-31771R1 

USA vs Russia in the scientific arena 

Dear Dr. Abramo:

I'm pleased to inform you that your manuscript has been deemed suitable for publication in PLOS ONE. Congratulations! Your manuscript is now with our production department. 

Kind regards, 

on behalf of

Dr. Bruno Miguel Pinto Damásio 

Academic Editor

PLOS ONE